# Evaluation of the preclinical analgesic efficacy of naturally derived, orally administered oil forms of Δ9-tetrahydrocannabinol (THC), cannabidiol (CBD), and their 1:1 combination

Katja Linher-Melville[1,2], Yong Fang Zhu[1,2], Jesse Sidhu[2], Natalka Parzei[2], Ayesha Shahid[2], Gireesh Seesankar[2], Danny Ma[2], Zhi Wang[2], Natalie Zacal[2], Manu Sharma[2], Vikas Parihar[3], Ramesh Zacharias[3], Gurmit Singh[1,2]*

1 Michael G. DeGroote Institute for Pain Research and Care, McMaster University, Hamilton, Ontario, Canada, 2 Department of Pathology & Molecular Medicine, McMaster University, Hamilton, Ontario, Canada, 3 Michael G. DeGroote Pain Clinic, McMaster University Medical Centre, Hamilton, Ontario, Canada

* gsingh@mcmaster.ca

## Abstract

Chronic neuropathic pain (NP) is a growing clinical problem for which effective treatments, aside from non-steroidal anti-inflammatory drugs and opioids, are lacking. Cannabinoids are emerging as potentially promising agents to manage neuroimmune effects associated with nociception. In particular, Δ9-tetrahydrocannabinol (THC), cannabidiol (CBD), and their combination are being considered as therapeutic alternatives for treatment of NP. This study aimed to examine whether sex affects long-term outcomes on persistent mechanical hypersensitivity 7 weeks after ceasing cannabinoid administration. Clinically relevant low doses of THC, CBD, and a 1:1 combination of THC:CBD extracts, in medium chain triglyceride (MCT) oil, were orally gavaged for 14 consecutive days to age-matched groups of male and female sexually mature Sprague Dawley rats. Treatments commenced one day after surgically inducing a pro-nociceptive state using a peripheral sciatic nerve cuff. The analgesic efficacy of each phytocannabinoid was assessed relative to MCT oil using hind paw mechanical behavioural testing once a week for 9 weeks. *In vivo* intracellular electrophysiology was recorded at endpoint to characterize soma threshold changes in primary afferent sensory neurons within dorsal root ganglia (DRG) innervated by the affected sciatic nerve. The thymus, spleen, and DRG were collected post-sacrifice and analyzed for long-term effects on markers associated with T lymphocytes at the RNA level using qPCR. Administration of cannabinoids, particularly the 1:1 combination of THC, elicited a sustained mechanical anti-hypersensitive effect in males with persistent peripheral NP, which corresponded to beneficial changes in myelinated Aβ mechanoreceptive fibers. Specific immune cell markers associated with T cell differentiation and pro-inflammatory cytokines, previously implicated in repair processes, were differentially up-regulated by cannabinoids in males treated with cannabinoids, but not in females, warranting further investigation into sexual dimorphisms that may underlie treatment outcomes.

**Data Availability Statement:** All relevant data are within the paper and its Supporting Information files.

**Funding:** KLM - Michael G. DeGroote Centre for Medicinal Cannabis Research, McMaster University G. Singh - Canadian Institutes of Health Research The funders had no role in study design, data collection and analysis, decision to publish, or preparation of the manuscript.

**Competing interests:** The authors have declared that no competing interests exist.

## Introduction

Neuropathic pain (NP) significantly affects the daily quality of life of patients suffering from this condition. Depending on its etiology, NP may involve both the peripheral and central nervous systems (PNS and CNS, respectively). In preclinical models representing surgically induced peripheral nerve damage, nociceptors undergo sensitization, and afferent fibres may develop patterns of ectopic neural discharge. As a consequence, a complex response in the spinal cord may be evoked that includes activated immune cells, ultimately resulting in neuronal hyper-excitation [1, 2]. The most effective therapeutics currently available and clinically recommended for the management of peripheral NP act by attenuating aberrant neural discharge-induced firing [1]. Initiating treatment at an early stage post-nerve injury (for example, directly after a surgery) may alter immune responses and limit or delay the development of persistent hypersensitivity, potentially blocking the propagation of peripheral impulses into the CNS. Indeed, early treatment initiation has been shown to delay the development of hyperalgesia in a rat model of NP [3]. As proposed by Neil and Macrae (2009), reducing nociceptive input into the spinal cord during and after surgery may be "sensible" to attenuate sensitization of the nervous system, and anaesthetic techniques that accomplish this aim should be associated with reduced post-operative as well as chronic pain [4]. In this context, research supports that local anaesthesia could provide potential benefits [5, 6]. However, it may not be practically feasible to entirely eliminate persistent pain using this approach [7], and it has therefore been suggested that a variety of different analgesic agents be administered around the time of surgery.

During the initiation and chronification of NP, neurons are affected by innate and adaptive immune responses that take place at the site of injury, the dorsal root ganglia (DRG), and the region of the spinal cord innervated by relevant afferent nociceptive fibers, which includes central microglial activation. The effects of cytokines and other immune cell-derived factors underlie many painful conditions that include unresolved inflammation [8]. Neutrophils play a key role [9], and T lymphocytes are emerging as another component of pain chronification [10, 11]. In patients with persistent pain, CD4+ T helper (Th) 1 (Th1) and Th2 cell imbalances have been reported [12, 13], and changes in the ratio of Th17 and T regulatory (Treg) cells may also occur [14]. Interestingly, various nerve constriction models using T cell-deficient rodents have demonstrated significant changes in nociceptive behaviours [14–16]. Numerous sex differences in peripheral immune cell profiles have been reported in rodents (reviewed in [17]) and in humans under homeostatic conditions, particularly in lymphocyte populations [18]. An important aspect in aberrant immune modulation of neurotransmission and the development of central sensitization may be the overall level of peripheral immune cell infiltration, which has been shown to be affected by sex differences in rodents [19]. Overall, sex differences may significantly influence pain perception, nociceptive processing, and neuroimmune responses [17].

While cannabinoids promote anti-nociception in acute preclinical models of pain [20] and may provide analgesia in patients with chronic NP, there remains a need for studies to evaluate parameters such as the dose, the duration of treatment, and the outcome on physiological and psychological functions [21] in the context of age, sex, and gender, which may significantly affect therapeutic outcomes. In addition, studies examining sustained anti-nociceptive effects following early cannabinoid administration in long-term preclinical pain models are scarce, particularly with regard to oral dosing regimens. The predominant components of phytocannabinoid extracts, Δ9-tetrahydrocannabinol (THC) and cannabidiol (CBD), are able to modulate nociceptive thresholds, and THC suppresses the activities of T lymphocytes [22, 23], with CBD also altering the functional activity of these cells [24–26]. THC interacts with endogenous receptors, including cannabinoid receptor 1 (CBR1) and 2 (CBR2), which are expressed by

neurons and immune cells. CBD does not appear to significantly associate with either of these G protein-coupled receptors, instead modulating the effects of THC and targeting the TRP family of ion channels [27]. This is relevant, given that certain classes of sensory neurons express the known nociceptor transient receptor potential vanilloid 1 (TRPV1), which is also expressed by T lymphocytes [28]. In addition, it is important to note that other receptors, such as GPR55, NMDAR and GABAR, are also modulated by cannabinoids [29], with CBD directly agonising the 5-HT1A receptor [30]. Therefore, mechanistically distinct anti-nociceptive effects may be elicited in a cannabinoid-dependent manner [31]. Combining THC and CBD at different ratios may be beneficial [32–34], with the 1:1 ratio of THC and CBD in oro-mucosal Sativex (GW Pharmaceuticals) showing promise in providing therapeutic relief for pain as well as allodynia in patients with peripherally originating NP [35].

The current study assessed the early treatment efficacy of low doses of orally administered natural cannabis oil extracts in male and female sexually mature Sprague Dawley rats with an impingement of the peripheral sciatic nerve. The rationale for orally administering naturally-extracted cannabis in MCT oil was that this particular formulation was used by patients at the Michael G. DeGroote Pain Clinic, McMaster University Medical Centre, as an alternative to smoking combusted cannabis or its vapor-based inhalation, not only to better control for dosing, but to avoid potential side-effects on lung health [36]. While the entourage effect of other constituents present in the plant-based extracts, such as terpenes and flavonoids [37, 38], cannot be dismissed, each formulation used in the current study was highly enriched for either THC or CBD (CannTrust). A day after surgical nerve cuff implantation, rats were orally gavaged for 14 consecutive days with either MCT oil alone, representing the vehicle control, or THC, CBD, or a clinically relevant 1:1 combination of the two cannabinoids. Effects on mechanical hypersensitivity were behaviourally assessed on a weekly basis for 9 weeks, and endpoint *in vivo* intracellular electrophysiology was applied to characterize sustained soma threshold changes in DRG associated with primary afferent sensory neurons affected by the sustained peripheral nerve constriction injury. The expression of CD4+ lymphocyte-associated markers was evaluated at the mRNA level in endpoint-isolated thymi, spleens, and lumbar 3 to 6 (L3-6) DRG. While this study was limited to specific clinically relevant low doses of THC (at a non-psychotropic concentration) and CBD oils, it appears that the early treatment efficacy of orally administered cannabinoids on persistent peripherally induced NP is dependent on sex. Males responded behaviourally and electrophysiologically to all three treatment regimens, with the most robust effects obtained with the combination of THC:CBD, while females failed to recover in response to any of the treatments. Significant sex differences were also apparent in thymic, splenic, and DRG marker expression levels, supporting an underlying dimorphism that affects neuroimmune cross-talk in this sustained NP condition.

## Materials and methods

Animal experiments were in keeping with the Guide to the Care and Use of Experimental Animals, Vol. 1 and 2, of the Canadian Council on Animal Care, with relevant protocols reviewed and approved by the McMaster University Animal Research Ethics Board. Work with cannabis was conducted under authorization of a Health Canada licence issued in accordance with the Cannabis Act and Cannabis Regulations. All animals were euthanized at experimental endpoint by decapitation according to the laboratory's Animal Utilization Protocol.

### Sciatic nerve cuff surgery

Upon arrival at the McMaster University Central Animal Facility, 170–200 gram (g) sexually mature male and female Sprague Dawley (SD) rats (Charles River Laboratories) underwent

acclimatization for one week, and were then pre-experimentally handled by dedicated research staff prior to baseline (BL) behavioural von Frey testing. A validated method based on surgically implanting a sciatic nerve cuff [39–42] was used to generate a peripheral neuropathic injury. Animals were anesthetised via intraperitoneal injection of a mixture of 0.1 mg/100 g acepromazine (Ayerst Veterinary Laboratories), 5 mg/100 g ketamine (Vetoquinol N.-A. Inc.), and 0.5 mg/100 g xylazine (Bayer Inc.). A mid-thigh incision into the muscle, exposing the sciatic nerve associated with the right rear limb, facilitated fitting of a longitudinally slit segment of 0.5 mm polyethylene-90 tubing (Fisher Scientific Ltd.) around the nerve (sciatic nerve "cuff"). The underlying incision was sutured, with the surrounding layer of skin being closed with surgical staples. Antibiotic ointment (0.2% nitrofurazone; Vetoquinol N.-A., Inc.) was applied to the wound in addition to subcutaneously injecting a solution of 0.01 ml/100 g of Baytril (Bayer Inc.). Sham surgeries were also carried out on separate groups of rats that underwent the same surgical procedures (incision and lifting of the sciatic nerve in the same location) without implantation of a cuff. Two independent experiments that included both male and female SD rats were carried out, with results pooled to ensure that adequate animal numbers were included for statistical relevance.

## Administration of cannabis oil

25 mg/ml of purified, concentrated THC or CBD-enriched extracts, suspended in medium chain triglyceride (MCT) oil (vehicle), were kindly provided by CannTrust. The concentration of CBD that was tested, based on the lowest limit of standard patient dose titration guidelines from CannTrust and the Michael G. DeGroote Pain Clinic (Hamilton Health Sciences), was 25 mg per day per adult human subject (taking into consideration an estimated average body weight of 60 kg). For THC, a non-psychotropic dose of 5 mg per day was selected, and for the 1:1 THC and CBD combination regimen, 12.5 mg of THC and 12.5 mg of CBD per day was used. To achieve an experimental rat body weight-adjusted equivalent dose, each cannabis stock solution was diluted in vehicle. For example, for a 200 g rat, the dose equivalent representing 25 mg of CBD was 0.0833 mg, and for 5 mg of THC, it was 0.0167 mg. The working concentration of each cannabinoid was freshly prepared and the administered volume adjusted daily for incremental increases in rat body weight for the 14-day treatment regimen, commencing one day after sciatic nerve cuff surgery. Each dose was administered via oral gavage using sterile two-inch, 18-gauge disposable, malleable animal feeding needles (Fisherbrand™ 102, catalogue number 01-208-88; Fisher Scientific Ltd.), at a final volume of 1 ml, at the same time each day (2–3 pm).

## Von Frey behavioural test

The research staff that carried out von Frey behavioural testing varied between experiment #1 and experiment #2, which were conducted approximately 6 months apart. However, within each given experiment, the same individuals handled all animals from one testing day to the next. Experimenters were blinded to treatment, with codes corresponding to the administered oil formulation ascribed to each cage, which were not shared with those conducting behavioural testing. Von Frey behavioral testing was carried out prior to surgery to obtain a BL value of mechanical paw withdrawal thresholds, and once a week thereafter for 9 weeks to track the development of tactile hypersensitivity characteristic of NP. Animals that did not exhibit a normal BL prior to nerve cuff surgery (corresponding to a von Frey score of 9 or higher) were removed from the study. The final number of males, combined from two independent experiments, was n = 10 for vehicle (VEH), and n = 9 for THC, CBD, and the 1:1 THC to CBD combination groups. For females, the final number of rats in each group pooled

from the two independent experiments was n = 11 for VEH, n = 10 for THC, n = 9 for CBD, and n = 10 for the 1:1 THC to CBD combination. Animals of each sex were randomly assigned into each group after BL von Frey assessments. Each rat was placed in a transparent Plexiglas box, the floor of which contained holes 0.5 cm in diameter spaced 1.5 cm apart to facilitate paw accessibility [40, 42, 43]. Animals were habituated to the box for 20 min until major cage exploration and grooming activities ceased. Von Frey filaments of different weights (Stoelting Co.) were applied, in ascending order, until a brisk foot withdrawal response was observed [44]. The mechanical withdrawal thresholds were determined using the up-down method [45], in which von Frey filaments were applied for 3–4 sec, repeated at 3 sec intervals for a total of 5 measurements, each at a different spot on the plantar surface of the right hind paw (corresponding to the limb bearing the sciatic nerve cuff or the sham-operated limb). Filaments were applied in ascending order of force (starting with the 2 gram filament) until a clear withdrawal response was observed. When this did not occur at a force corresponding to 2 grams, the next-heaviest filament was applied, with this process continued until a 50% withdrawal response threshold was achieved. Brisk foot withdrawal in response to the mechanical stimulus was interpreted to be a valid response. To minimize the effect of the circadian cycle, all behavioral assessments were performed at approximately the same time each day (starting at 11 am in experiment 1 and 2 pm for experiment 2).

### *In vivo* intracellular DRG recordings

The acute intracellular electrophysiological recording techniques have been reported previously in animal models of NP [40, 41, 43, 46] and cancer pain [47, 48]. After the last behavioural test was performed during week 9 (day 63 post-nerve cuff surgery), each rat was initially anesthetised with an intraperitoneally delivered mixture of acepromazine, ketamine, and xylazine. To facilitate intravenous drug infusion, the right jugular vein was cannulated, and each rat was positioned into a stereotaxic frame by rigidly clamping the spinal cord at L2 and L6. The L4 DRG was selected for study, as it contains the highest number of afferent somas associated with the region innervated by the sciatic cuff-bearing hind limb. The laminectomized L4 dorsal root was placed on a bipolar electrode (FHC). To prevent drying, the exposed spinal cord and DRG were covered with 37˚C-preheated paraffin oil. Body temperature was rectally monitored and maintained at 37˚C using infrared heat.

During recordings, 20 mg/kg sodium pentobarbital (Ceva Sante Animal) was applied to maintain surgical anesthesia. A tracheal cannula ensured mechanical ventilation of each rat via a Model 683 Harvard Ventilator (Harvard Apparatus), with CapStar-100 End-Tidal $CO_2$ analyzer (CWE) parameters adjusted to 40–50 mmHg. Immediately prior to commencing electrophysiological recordings, an initial 1 mg/kg dose of pancuronium (Sandoz) was administered to the animal. Its effects were periodically allowed to wear off in order to confirm a surgical level of anesthesia, which was continuously monitored by observing pupil diameter and responses to a noxious forepaw pinch. Additional sodium pentobarbital and pancuronium were administered as needed, approximately every hour, at 1/3 of the previous dose via the jugular cannula.

Intracellular recordings from somas in the exposed DRG were performed using microelectrodes prepared from borosilicate glass (1.2 and 0.68 mm in outside and inside diameters, respectively; Harvard Apparatus), which were pulled using a Brown-Flaming Model P-87 pipette puller (Sutter Instrument Co.) and filled with 3 M KCl (DC resistance of 50–70 MΩ). Signals were recorded (Multiclamp 700B amplifier, Molecular Devices) and digitized on-line (Digidata 1322A interface, Molecular Devices) with pClamp 9.2 software (Molecular Devices). An IW-800 micromanipulator (EXFO) was used to advance the microelectrode in 2 μm

increments until an abrupt hyperpolarization corresponding to at least 40 mV was obtained. Upon confirmation of a stable membrane potential, a single stimulus was applied to the dorsal root, provoking an action potential (AP). The Protocol Editor function (pClamp 9.2 software) was then used to evoke a somatic AP via stimulation with a single rectangular intracellular depolarizing voltage pulse.

## DRG neuron excitability

Soma excitability was measured by evoking APs in the somata of a DRG neuron by direct depolarizing current injection [40]. To quantify this excitability, the threshold of depolarizing current pulses of 100 ms each, injected with an amplitude of 500 to 4000 pA in 500 pA increments into the soma was determined using the Protocol Editor function in pClamp 9.2. The conduction velocity (CV) was used to categorize DRG sensory neurons as follows: C-fiber neurons ≤0.8 mm/ms, Aδ-fiber neurons 1.5–6.5 mm/ms, and Aβ-fiber neurons >6.5 mm/ms [41, 49]. The threshold of activation, the depth of the receptive field, and the pattern of adaption were the major factors used to further classify neurons into LTM, HTM, and unresponsive neurons [41, 49–51]. HTM neurons responded to noxious stimuli, including a noxious pinch and application of pointed objects such as the sharp end of a syringe needle, whereas LTM neurons responded to innocuous stimuli such as a moving brush, light pressure with a blunt object, a light manual tap, or vibration. Other major factors were used to further classify Aβ LTM neurons as either cutaneous (CUT) or slowly adapting muscle spindle (MS) neurons. To further specify, a neuron was classified as a MS afferent neuron if it could be activated by touching along the muscle belly or changing joint position, and was also not activated by touch or pressure stimuli that were only applied to the covering skin; in this latter case the skin was lifted or pulled aside to ensure that stimuli were not applied to deeper tissue. An MS neuron also had to have a subcutaneous receptive field, which was confirmed by thorough receptive field testing and response to low intensity stimulation of deep structures.

## RNA extraction and qPCR analysis

**Thymi and spleens.** The thymus is the primary lymphoid organ in which T cells undergo differentiation and maturation. It is comprised of bone marrow-derived hematopoietic stem cells that differentiate into naïve thymocytes (T0 cells), as well as dendritic and epithelial cells that regulate the functional selection of a self-tolerant T cell repertoire to prevent autoimmune responses. The latter may be of particular importance, given that autoimmune-like processes may contribute to pain chronification, and autoimmune disease is more prevalent in women than in men [52]. Therefore, T lymphocyte-associated marker profiles in male and female thymi rapidly dissected from decapitated rats at the 9-week experimental endpoint were evaluated. In addition, the spleens from these same animals were also collected, and each tissue was snap-frozen in liquid nitrogen and stored at -80°C, with RNA extracted using Trizol (Invitrogen). Of note, attention was given to the anatomy of the spleen in choosing a representative piece of tissue for RNA extraction, given that its red pulp regions contain red blood cells, while white pulp contains lymphoid follicles representing B cell-enriched areas and periarteriolar lymphoid sheaths containing T cells. Depending on the portion of the spleen that was selected for RNA extraction, the splenic red or white pulp regions could be differently represented. This is not the case for the thymus, since its organization is more homogeneous and the majority of the organ, based on its considerably smaller size, was used for RNA extraction purposes in the current study. The total number of thymi and spleens isolated from individual male and female rats per treatment group that were analyzed at the mRNA level is summarized in Table 1. Following treatment with DNase I (Ambion), cDNA was prepared from 5 μg of total

**Table 1. Number (n) of individual animals used for RNA extraction and qPCR.**

| Sex | Treatment | Thymus (n) | Spleen (n) | DRG (n) |
|---|---|---|---|---|
| Male | Sham | 2 | 2 | 2 |
| | VEH | 7 | 7 | 3 |
| | CBD | 6 | 6 | 3 |
| | THC | 7 | 7 | 3 |
| | 1:1 Combination | 7 | 7 | 3 |
| Female | Sham | 3 | 3 | 3 |
| | VEH | 5 | 5 | 3 |
| | CBD | 9 | 9 | 3 |
| | THC | 7 | 7 | 3 |
| | 1:1 Combination | 6 | 6 | 3 |

RNA using the SuperScript III kit (Invitrogen). qPCR was carried out using SYBR Green (BioRad) and rat-specific primers listed in Table 2. The integrity of each product was verified by size using agarose gel electrophoresis coupled with an examination of the specific melt peak. Relative mRNA levels were calculated using the 2-delta delta Ct method [53] following primer validation.

**L3-L6 pooled DRG.** At endpoint, the spinal cord of each rat was carefully dissected, immediately submerged into excess RNALater solution (Ambion), and stored at -20˚C for one week. DRG associated with L3-L6 on the right side of the animal, each innervated by afferents from the cuff-bearing sciatic nerve, were carefully excised under a stereoscope. L3-L6 DRG were pooled for individual rats and RNA was extracted through mechanical homogenization in lysis buffer and purified using the Nucleospin RNA XS kit (Macherey-Nagel). The total number of DRG isolated from individual male and female rats per treatment group that were analyzed at the mRNA level is summarized in Table 1. RT-qPCR was then carried out as described.

## Data analysis and statistics

GraphPad Prism software (GraphPad Software, Inc.) was used for all statistical analyses and graphing. von Frey behavioural data is presented as the mean gram force ± SEM, with statistical significance assessed using a 2-way ANOVA with a Dunnett's post hoc analysis to compare each group to its BL. A 1-way ANOVA with a Tukey's post-hoc test was carried out for

**Table 2. Rat-specific primers used to detect and quantify markers associated with different CD4+ lymphocyte subtypes at the mRNA level by qPCR, with reference primers listed below.**

| CD4+ T Subtype | Target | Primer Sequence (5' to 3') | | Product Size (bp) |
|---|---|---|---|---|
| | | Forward | Reverse | |
| Th1 | Ifng | TGCATTCATGAGCATCGCCA | CACCGACTCCTTTTCCGCT | 132 |
| | Tnfa | CCAGGAGAAAGTCAGCCTCCT | TCATACCAGGGCTTGAGCTCA | 87 |
| Th1, Th2, Treg | Il2 | CCCTGCAAAGGAAACACAGC | CTGCAGAGCTCCGAGTTCAT | 187 |
| Th2 | Il4 | TGCACCGAGATGTTTGTACC | GGATGCTTTTTAGGCTTTCC | 277 |
| | Il10 (Th1) | GCAGGACTTTAAGGGTTACTTGG | GGGGAGAAATCGATGACAGC | 181 |
| Th17 | Il17a | CCTTCTGTGATCTGGGAGGC | CACCAGCATCTTCTCCACCC | 165 |
| Reference | | Primer Sequence (5' to 3') | | Product Size (bp) |
| | | Forward | Reverse | |
| βActin | | CTGGCTCCTAGCACCATGAA | AAACGCAGCTCAGTAACAGTC | 195 |

**Table 3. Male von Frey behavioural data presented as the mean withdrawal force, in grams (g).**

| Time | Mean±SEM (g) | | | | |
|---|---|---|---|---|---|
| | Sham | VEH | CBD | THC | 1:1 Combination |
| BL | 13.7±1.3 | 14.6±0.4 | 14.2±0.6 | 13.4±1.1 | 14.8±0.2 |
| WK1 | 13.0±2.9 | 7.3±0.9 | 13.2±1.4 | 13.3±1.7 | 9.2±2.2 |
| WK2 | 15.7±2.4 | 7.1±1.9 | 13.7±2.0 | 11.1±2.0 | 11.3±2.6 |
| WK3 | 13.8±3.5 | 6.5±1.0 | 10.6±2.9 | 8.9±2.3 | 15.3±2.4* |
| WK4 | 17.4±3.4 | 6.1±1.1 | 15.4±2.3** | 12.2±2.6 | 13.2±2.5 |
| WK5 | 21.6±2.3 | 5.6±1.7 | 15.7±2.6** | 13.6±3.2* | 17.0±2.9*** |
| WK6 | 12.9±3.8 | 6.8±1.9 | 17.0±1.9** | 16.6±3.5** | 18.5±2.3** |
| WK9 | 15.8±5.0 | 6.4±2.3 | 20.1±2.9**** | 12.2±2.1 | 20.7±3.8*** |

Values with an asterisk (*) represent statistical differences between the VEH group and each of the individual treatment regimens at each time point over the course of the 9 week experiment, as determined by a 2-way ANOVA with a Bonferroni multiple comparison test (*$P<0.05$, **$P<0.01$, ***$P<0.001$, and ****$P<0.0001$).

comparisons at a specific experimental time point, with letters denoting significant differences between groups. In addition, a 2-way ANOVA with a Bonferroni's multiple comparison test was used to compare behavioural responses of males relative to females for each treatment across the time course. A P value of less than 0.05 ($P<0.05$) represents a significant difference between groups. Given that male and female rats gained weight at differing rates, with considerably different endpoint body masses, this variable was taken into account to allow for a comparison between the sexes for the von Frey behavioural test. Non-parametric electrophysiological data is presented as the mean threshold in nA ± SEM and was analyzed using either a Kruskal-Wallis analysis with the Dunn's multiple comparisons test or a Mann-Whitney t-test to compare between naïve and VEH-treated groups within each sex. $P<0.05$ was considered to indicate a significant difference. qPCR results are presented as mRNA levels relative to the mean of male sham animals at endpoint, with statistical significance analyzed by 1-way ANOVA with a Tukey's post-hoc comparison.

## Results

### Von Frey test

Von Frey data is summarized in Table 3 for males and Table 4 for females, which also list any specific significant differences within a sex between the VEH and each cannabinoid treatment

**Table 4. Female von Frey behavioural data presented as the mean withdrawal force, in grams (g).**

| Time | Mean±SEM (g) | | | | |
|---|---|---|---|---|---|
| | Sham | VEH | CBD | THC | 1:1 Combination |
| BL | 15.0±0.0 | 14.6±0.4 | 14.3±0.7 | 15.0±0.0 | 14.7±0.3 |
| WK1 | 16.2±1.7 | 9.2±2.0 | 11.5±1.8 | 9.7±1.9 | 10.6±2.0 |
| WK2 | 15.9±2.2 | 11.9±2.2 | 7.0±1.8 | 9.6±1.8 | 12.2±2.4 |
| WK3 | 15.0±3.4 | 7.7±2.2 | 11.5±2.7 | 6.8±2.3 | 9.3±1.9 |
| WK4 | 15.9±2.7 | 8.9±2.0 | 3.7±1.3 | 6.9±2.1 | 6.4±1.1 |
| WK5 | 12.0±4.1 | 5.0±1.2 | 5.9±1.6 | 6.9±2.5 | 11.0±2.4 |
| WK6 | 15.2±3.4 | 6.7±1.5 | 5.0±1.8 | 2.4±0.4 | 7.3±1.7 |
| WK9 | 11.4±2.7 | 7.8±2.1 | 4.8±0.9 | 5.7±1.6 | 5.3±1.5 |

Values with an asterisk (*) represent statistical differences between the VEH group and each of the individual treatment regimens at each time point over the course of the 9 week experiment, as determined by a 2-way ANOVA with a Bonferroni multiple comparison test (*$P<0.05$, **$P<0.01$, ***$P<0.001$, and ****$P<0.0001$).

at each time point. Measurements collected after the first week (WK1) post-incision without nerve cuff implantation (sham) revealed that neither males nor females differed significantly from their respective pre-surgical BL values, nor did they differ from each other at any week within the time course (Fig 1A; F = 0.9910 and P = 0.4448). In contrast, behavioural testing collected at WK1 post-nerve cuff surgery demonstrated that the male VEH-treated group experienced a significant 50% decrease in mechanical withdrawal threshold relative to pre-surgical BL values (Fig 1B; male BL: 14.6±0.4 g, WK1: 7.3±0.9 g, P<0.015). VEH-treated females exhibited a statistically significant change in withdrawal force at WK3, represented by a 47% decrease relative to female BL values (Fig 1B; female BL: 14.6±0.4 g, WK3: 7.7±2.2 g, P<0.01). This indicates successful induction of the NP model in both sexes, with mechanical hypersensitivity maintained throughout the duration of the 9 week experiment, with no statistical difference between the von Frey comparison of the male and female VEH-treated groups (F = 0.5469 and P = 0.7977).

A comparison for each of the three cannabis oil treatment regimens revealed that in males, THC, CBD, and the 1:1 combination of THC and CBD behaviourally alleviated persistent mechanical hypersensitivity (Fig 1C), as none of the treatments differed significantly from male pre-surgical BL values at any experimental time point (chart, Fig 1C). This effect was not observed in females for any of the treatments (Fig 1D). In addition to data plotted over the entire experimental time course, WK3 and WK6 data is represented in individual bar graphs, as these time points reflect the animals' behavioural responses one week and one month, respectively, after daily treatment had ceased (Fig 1E and 1F). A time point-focused 1-way ANOVA analysis carried out on WK3 male data revealed that of the cannabinoid treatments, only the 1:1 combination returned nociceptive behaviours to BL one week after ceasing daily gavage (Fig 1E, left panel; F = 7.843 and P<0.0001). By WK6, all three treatments did not differ significantly from BL (Fig 1E, right panel; F = 6.800 ad P<0.0001). A closer examination of the female data at WK3 revealed that only the behaviour of the CBD-treated group differed significantly from VEH-treated females by reflecting a return to BL (Fig 1F, left panel; F = 9.364 and P<0.0001), with none of the treatments alleviating mechanical hypersensitivity at WK6 (Fig 1F, right panel; F = 50.29 and P<0.0001).

Data in Fig 1C and 1D was replotted in Fig 2 to examine sex differences for each cannabinoid treatment regimen across the 9-week time course. A comparison of the THC effect between the sexes (F = 2.482 and P = 0.0198) showed a significant difference at WK6 (Fig 2A). At this time point, THC-treated males had a mean withdrawal score of 16.6±3.5 g compared to similarly treated females with a mean score of 2.4±0.4 g (P<0.0001). A sex difference was also statistically determined for treatment with CBD (F = 4.985 and P<0.0001), with specific significant differences between males and females at WK4 (P<0.001), 5 (P<0.01), 6, (P<0.001), and 9 (P<0.0001) (Fig 2B), corresponding to approximate 3 to 4-fold higher mechanical withdrawal force thresholds in treated males than in females. An analysis of the 1:1 combination treatment also produced significant sex differences (F = 3.764 and P = 0.0009), specifically at WK6 (P<0.01) and 9 (P<0.0001) (Fig 2C), with an approximately 3-fold higher paw withdrawal force in males compared to females treated with this cannabinoid oil regimen. Taken together, this data supports that sex differences affect the efficacy of these oils at the doses selected for evaluation.

## Electrophysiology

The activation thresholds of sensory neurons in response to intracellular current injection were evaluated at the 9-week experimental endpoint to determine whether there were any sustained differences in DRG soma excitability in response to cannabis oil administration, which

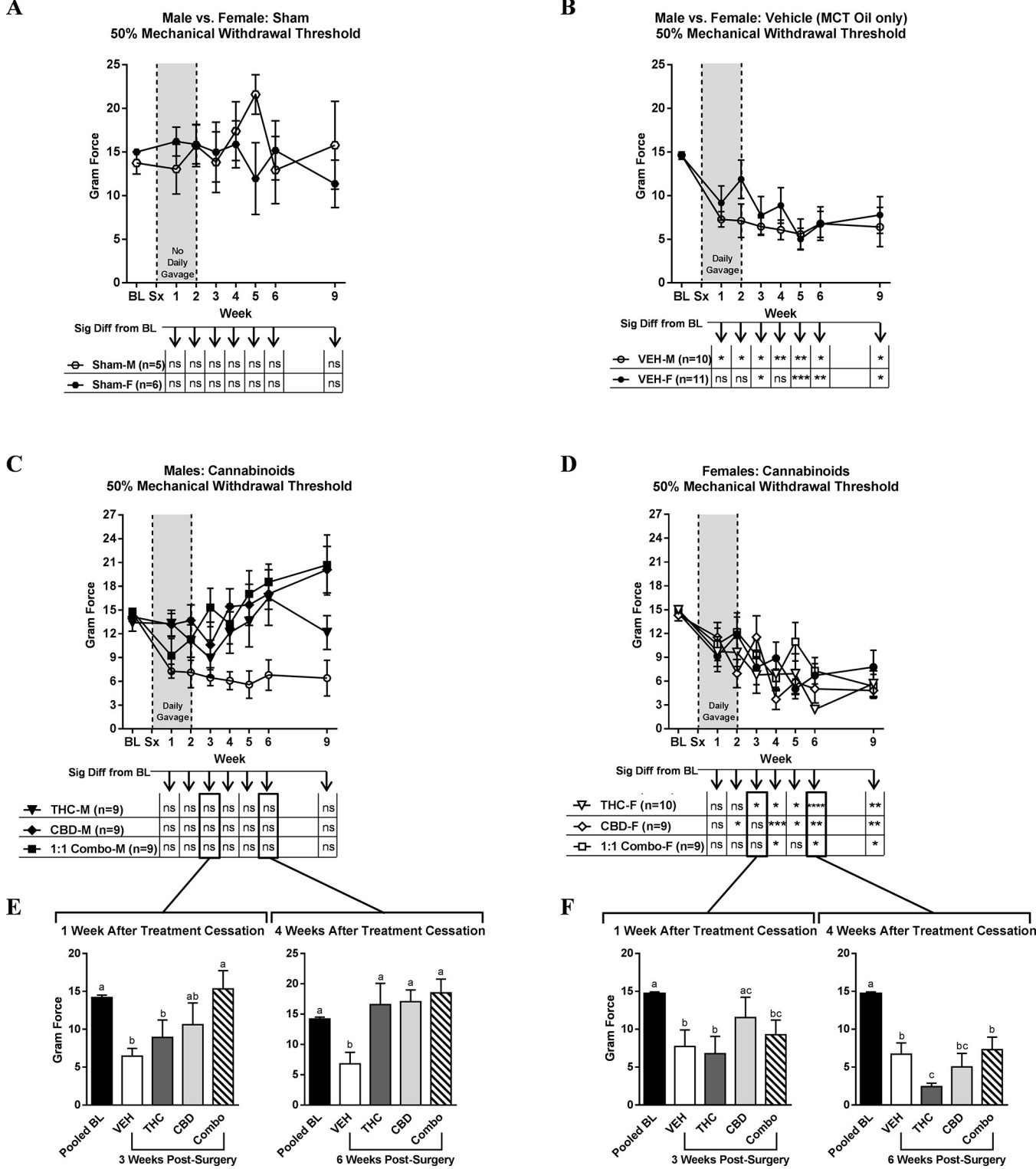

**Fig 1. Mechanical withdrawal thresholds for behavioural comparisons of sham, vehicle-treated, and cannabinoid-treated males and females.** Withdrawal thresholds plotted over a 9-week time course, with baseline (BL) and surgical intervention (Sx) indicated in **A)** male and female sham rats that underwent surgery without implantation of a sciatic nerve cuff; **B)** sciatic nerve cuff-implanted male and female rats treated with MCT oil alone (vehicle; VEH); **C)** sciatic nerve-cuff implanted male rats treated with THC, CBD, or a 1:1 combination of THC:CBD (Combo) for 2 weeks compared to VEH; **D)** sciatic nerve-cuff implanted female rats treated with THC, CBD, or a 1:1 Combo for 2 weeks compared to VEH. Data in **E)** and **F)** reflect time points corresponding to 1 week and 1 month after treatments had ceased (3 and 6 weeks post-nerve cuff implantation in **C)** and **D)**), respectively. All data is expressed as the mean gram

force ± SEM. In **A)**, **B)**, **C)**, and **D)**, 2-way ANOVA with a Dunnett's post hoc analysis was used. Stars (*) in the charts beneath each graph indicate the level of significant differences relative to BL (*P<0.05, **P<0.01, ***P<0.001, ****P<0.0001). A 1-way ANOVA with a Tukey's post-hoc test was carried out for comparisons at each of the specific experimental time points in **E)** and **F)**, with letters denoting differences between groups. The number of rats per group is indicated in each chart.

had ceased 7 weeks prior to electrophysiological recordings. The exact activation thresholds are listed in Tables 5 and 6 for males and females, respectively, with results graphically summarized in Fig 3. The parameters used are shown in Fig 3A. Discharge was evoked by injecting a

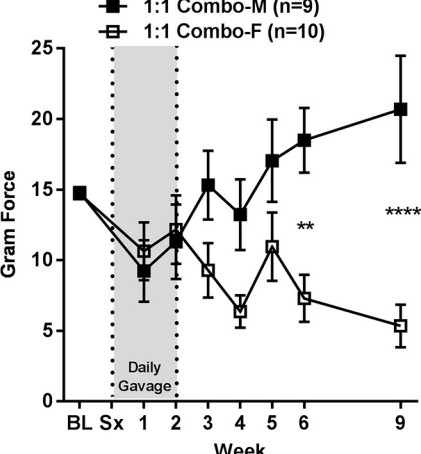

**A**

**Male vs. Female: THC**
**50% Mechanical Withdrawal Threshold**

▼ THC-M (n=9)
△ THC-F (n=9)

****

**B**

**Male vs. Female: CBD**
**50% Mechanical Withdrawal Threshold**

◆ CBD-M (n=9)
◇ CBD-F (n=9)

*** ** *** ****

**C**

**Male vs. Female: 1:1 Combination**
**50% Mechanical Withdrawal Threshold**

■ 1:1 Combo-M (n=9)
□ 1:1 Combo-F (n=10)

** ****

**Fig 2. Mechanical withdrawal thresholds for behavioural comparisons between males and females treated with cannabinoids.** Withdrawal thresholds plotted over a 9-week time course, with baseline (BL) and surgical intervention (Sx) indicated, in sciatic nerve-cuff implanted males and females treated for 2 weeks with **A)** THC; **B)** CBD; **C)** a 1:1 combination of THC:CBD (Combo). All data is expressed as the mean gram force ± SEM. A 2-way ANOVA with a Bonferroni's multiple comparison test was used to compare behavioural responses of males relative to females for each individual treatment across the time course. Stars (*) indicate the level of significant differences at indicated time points (*P<0.05, **P<0.01, ***P<0.001, ****P<0.0001). The number of rats per group is indicated. Data shown here to highlight sex differences in a specific treatment response are also plotted in Fig 1C and 1D.

series of depolarizing current pulses into DRG soma through the recording electrode, with representative raw recordings shown for male Aβ HTM neurons of the 1:1 THC to CBD combination-treated group (Fig 3B). All comparisons were carried out relative to naïve animals (no surgical intervention) for each sex. The excitability of male and female C HTM neurons did not significantly differ between naïve animals and VEH-treated nerve-cuff implanted counterparts, or across any of the treatment groups relative to VEH within each sex (Fig 3C). In contrast, the activation threshold of male Aβ HTM neurons of nerve cuff-bearing VEH-treated rats underwent a significant decrease in their activation threshold compared to naïve males (P<0.01). In addition, the male THC- and Combo-treated groups within this class of neurons exhibited significantly increased thresholds relative to VEH-treated rats (Fig 3D, upper panel; P<0.01). For female Aβ HTM neurons, a Mann-Whitney non-parametric t-test revealed that naïve animals had a significantly higher threshold than nerve-injured VEH-treated rats (P<0.05), and only those treated with THC showed a significant increase in threshold activity relative to VEH-treated counterparts in the female multiple comparison (Fig 3D, lower panel; P<0.001). The activation threshold of male CUT neurons did not undergo a change in nerve-injured compared to naïve rats (Fig 3E, upper panel). In THC- and Combo-treated males, the threshold was significantly increased relative to VEH-treated counterparts (Fig 3E, upper panel; P<0.05 and P<0.001, respectively). In addition, there was a statistically significant increase between naïve CUT neurons and Combo-treated males (P<0.01), and CBD- and Combo-treated males (P<0.05). No statistically significant differences were obtained for female CUT neurons, irrespective of injury state or treatment (Fig 3E, lower panel). The threshold of male MS neurons reflected a similar recording profile as was obtained for Aβ HTM neurons in males (Fig 3F, upper panel). Specifically, via a Mann-Whitney t-test, the nerve cuff-bearing VEH-treated group reflected a significantly lower threshold relative to naïve MS nerves (P = 0.02), while the THC and Combo groups showed a return to a naïve profile for this class of neurons in the multiple comparison (Fig 3F, upper panel; P<0.05 and P<0.01, respectively). In marked contrast, no differences were observed between any of the female groups for MS neurons (Fig 3F, lower panel). Of final note, a cross-sex comparison did not reveal any significant differences between male and female sham, VEH, THC, CBD, or Combo groups for any of the fiber classes, with the exception of the CUT combo (P = 0.04), which had a higher threshold in males than females (2.81 ± 0.21 vs. 2.06 ± 0.24 nA).

## Thymic male and female CD4+ T cell-associated marker expression

The mRNA level of a repertoire of markers of CD4+ T cells, selected primarily to represent Th1, Th2, and Th17 cells, were assessed in the thymus, as this class of lymphocytes has been reported to be present in the DRG following nerve injury [54]. Results of thymic qPCR in endpoint-collected tissues are graphically summarized in the left panel of Fig 4. In sham-operated rats, *Il2* mRNA levels were significantly higher in females than males (Fig 4A; P = 0.0004, t-test). In males and females with a nerve cuff that had been treated with VEH, thymic *Il2* mRNA levels were significantly lower relative to their respective sham counterparts, with THC, CBD, and the 1:1 combination having significantly increased the expression of this cytokine, returning it to sham levels, irrespective of sex (Fig 4A; Male 1-way ANOVA: F = 7.869 and P = 0.0002; Female 1-way ANOVA: F = 7.511 and P = 0.0003). While *Ifng* levels were not significantly different between the male and female sham-operated groups (Fig 4B; P = 0.1033, t-test), *Tnfa* mRNA was significantly higher in female compared to male shams (Fig 4C; P = 0.0004, t-test). Male thymic *Ifng* (Fig 4B; Male 1-way ANOVA: F = 14.15 and P<0.0001) and *Tnfa* (Fig 4C: Male 1-way ANOVA: F = 11.25 and P<0.0001) levels followed the same pattern of changes that were established for the male *Il2* analysis. However, female thymic *Ifng*

**Table 5. Male activation thresholds of sensory neurons in response to intracellular current injection tested to determine differences in soma excitability in response to cannabinoid treatment.**

| Treatment | Threshold (nA) | | | |
|---|---|---|---|---|
| | **C HTM** | **Aβ HTM** | **CUT** | **MS** |
| **Naive** | 1.92 ± 0.28, n = 13 | 1.98 ± 0.14, n = 20 | 1.62 ± 0.17, n = 17 | 1.04 ±0.16, n = 14 |
| **VEH** | 1.45 ± 0.16, n = 10 | 1.06 ± 0.15, n = 18 | 1.34 ± 0.19, n = 16 | 0.57 ± 0.05, n = 14 |
| **THC** | 1.82 ± 0.22, n = 14 | 2.59 ± 0.56, n = 11 | 2.35 ± 0.22, n = 13 | 1.25 ± 0.23, n = 12 |
| **CBD** | 1.89 ± 0.32, n = 9 | 1.75 ± 0.23, n = 12 | 1.60 ± 0.19, n = 10 | 0.81 ± 0.07, n = 13 |
| **1:1 Combination** | 2.07 ± 0.35, n = 14 | 2.43 ± 0.39, n = 14 | 2.81 ± 0.21, n = 13 | 1.15 ± 0.13, n = 13 |

levels did not significantly change between the sham, VEH, and cannabinoid treatment groups (Fig 4B; Female 1-way ANOVA: F = 2.217 and P = 0.0918). For *Tnfa*, mRNA levels significantly decreased between the female sham and nerve-cuff implanted groups, although none of the cannabinoid treatments changed its levels relative to VEH (Fig 4C; female 1-way ANOVA: F = 14.25 and P<0.0001). The thymic expression of *Il17a* did not significantly differ between female and male shams (Fig 4D; P = 0.6379, t-test). Male thymic *Il17a* (Fig 4D; Male 1-way ANOVA: F = 5.301 and P<0.0028) levels followed the same pattern of changes that were established for the male *Il2* analysis. In contrast, female thymic *Il17a* levels did not significantly change between the sham, VEH, and any of the cannabinoid treatment groups (Fig 4D; Female 1-way ANOVA: F = 0.5823 and P = 0.6778). Results for representative markers of Th2 cells, *Il4* and *Il10*, are shown in S1A and S1B Fig, respectively.

## Splenic male and female CD4+ T cell-associated marker expression

A graphical summary of splenic qPCR results in endpoint-collected tissues is shown in the middle panel of Fig 4. An examination of splenic sham *Il2* (Fig 4A; P = 0.0412, t-test), *Ifng* (Fig 4B; P = 0.2419, t-test), and *Tnfa* (Fig 4C; P = 0.0828, t-test) mRNA levels revealed that only *Il2* levels differed significantly between the sexes, being higher in female relative to male sham-operated animals. The *Il2* mRNA level did not fluctuate significantly in males across any of the groups (Fig 4A; male 1-way ANOVA: F = 0.6713 and P = 0.6176). In contrast, the splenic *Il2* expression profile for the female comparison mirrored results obtained in the thymus for this T cell marker, with nerve cuff-implanted VEH-treated rats having a significantly lower *Il2* mRNA level compared to shams, and treatment with THC, CBD, and the 1:1 combination of both cannabinoids sex-specifically returning the expression of *Il2* to the level obtained in the female sham group (Fig 4A; female 1-way ANOVA: F = 4.823 and P = 0.0044). Splenic *Ifng* levels were not significantly affected by the presence of a nerve cuff in males relative to sham-operated counterparts, with cannabinoid treatments having no major effects in males (Fig 4B; male 1-way ANOVA: F = 3.296 and P = 0.0241), while in females, none of the groups differed

**Table 6. Female activation thresholds of sensory neurons in response to intracellular current injection tested to determine differences in soma excitability in response to cannabinoid treatment.**

| Treatment | Threshold (nA) | | | |
|---|---|---|---|---|
| | **C HTM** | **Aβ HTM** | **CUT** | **MS** |
| **Naive** | 2.12 ± 0.30, n = 13 | 2.06 ± 0.22, n = 18 | 1.27 ± 0.20, n = 11 | 0.81 ± 0.09, n = 8 |
| **VEH** | 1.80 ± 0.28, n = 10 | 1.37 ± 0.16, n = 15 | 1.68 ± 0.31, n = 11 | 0.72 ± 0.09, n = 9 |
| **THC** | 2.44 ±0.52, n = 8 | 3.30 ± 0.41, n = 10 | 2.33 ± 0.31, n = 9 | 0.94 ± 0.18, n = 8 |
| **CBD** | 1.94 ± 0.35, n = 8 | 1.88 ± 0.20, n = 12 | 2.59 ±0.50, n = 11 | 0.75 ± 0.09, n = 8 |
| **1:1 Combination** | 2.00 ± 0.27, n = 8 | 2.00 ± 0.38, n = 10 | 2.06 ± 0.24, n = 9 | 0.94 ± 0.15, n = 9 |

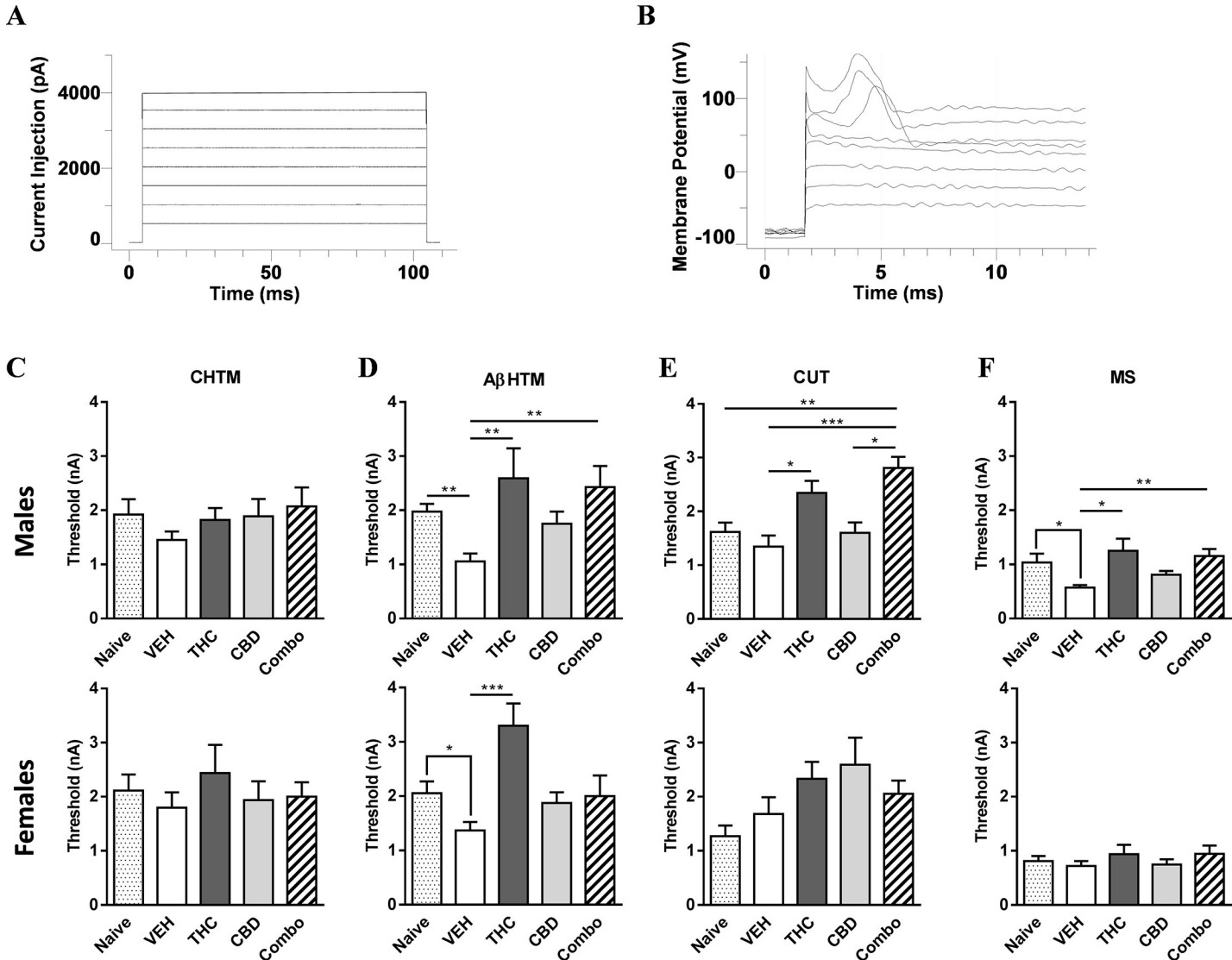

**Fig 3. A comparison of the *in vivo* activation threshold of sensory neurons of nerve cuff-bearing rats in response to early cannabinoid treatment relative to naïve animals.** Electrophysiological recordings were captured seven weeks after cannabinoid administration had ceased. **A)** The activation threshold was measured by evoking action potentials (AP) in the soma of L4 DRG neurons using stimulation by direct injection of depolarizing current; X-axis: Time (ms), Y-axis: current (pA). **B)** Representative raw recordings performed on male Aβ HTM neurons in the 1:1 THC to CBD combination (Combo)-treated group. X-axis: Time (ms), Y-axis: Voltage (mV). The threshold of **C)** C HTM, **D)** Aβ HTM, **E)** Aβ LTM CUT, and **F)** Aβ LTM MS fibers, recorded in males and females. **C-F)** The current threshold was defined as the minimum current required to evoke an AP by intracellular injection. Data are expressed as the mean threshold ± SEM for each treatment group. Differences between treatment groups were compared using the Kruskal-Wallis test with a Dunn's multiple comparison post hoc test; stars (*) with a corresponding line between groups indicate the level of significance (*$P<0.05$, **$P<0.01$, ***$P<0.001$). A bracket indicates significance determined via a Mann-Whitney t-test to compare between naïve and VEH-treated groups within a given sex (*$P<0.05$). HTM: high threshold mechanoreceptor, LTM: low threshold mechanoreceptor, CUT: Aβ LTM cutaneous neuron, MS: Aβ LTM muscle spindle neuron.

significantly from each other with regard to *Ifng* expression (Fig 4B; female 1-way ANOVA: F = 2.501 and P = 0.0650). Splenic *Tnfa* mRNA levels did not significantly change across any of groups in males (Fig 4C; male 1-way ANOVA: F = 0.7635, P = 0.5576). In females, while there was a significant drop in splenic *Tnfa* mRNA levels in the VEH-treated nerve cuff group compared to sham-operated counterparts, none of the cannabinoid treatments differed in *Tnfa* expression compared to VEH (Fig 4C; female 1-way ANOVA: F = 6.449 and P = 0.008).

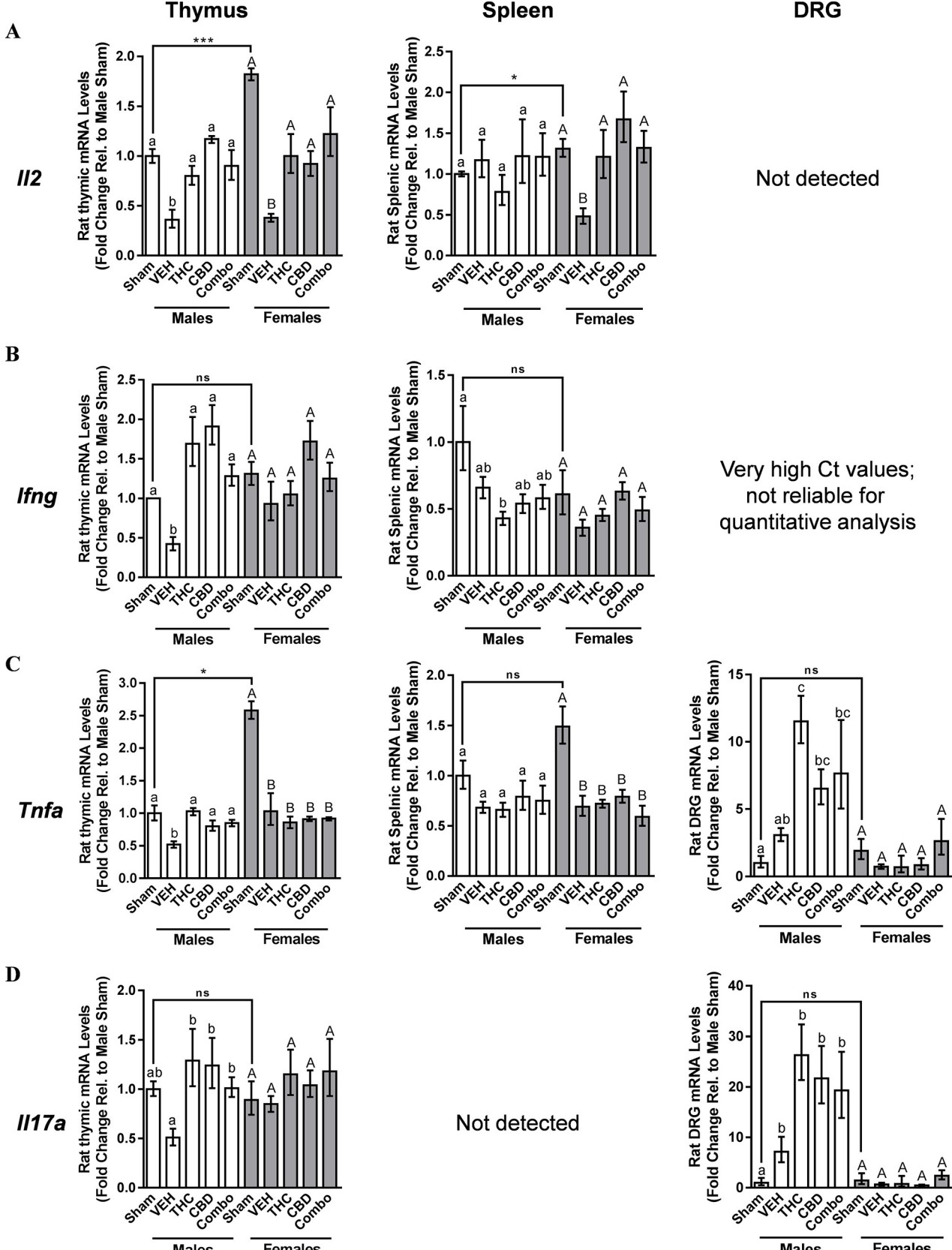

**Fig 4. A comparison of the CD4+ lymphocyte-associated marker profile in endpoint-collected sham, vehicle-treated, and cannabinoid-treated male and female thymi, spleens, and DRG.** RT-qPCR was carried out on mRNA extracted from 9-week endpoint-collected tissues of male and female rats that had been implanted with a sciatic nerve cuff, seven weeks after cannabinoid administration had ceased. Left, middle, and right panels represent data obtained from the thymus, spleen, and DRG, respectively, with results shown for **A)** *Il2*, **B)** *Ifng*, **C)** *Tnfa*, and **D)** *Il17a*. Data are expressed as the mean ± SEM for each treatment group, and is presented relative to male sham-treated rats, with statistical significance analyzed by 1-way ANOVA with a Tukey's post-hoc comparison. Different letters (male comparison: a, b, etc.; female comparison: A, B, etc.) represent differences between groups (P<0.05). For inter-sex comparisons between the sham groups, a t-test was applied, with significant differences indicated by stars (*). VEH: vehicle (MCT oil), Combo: 1:1 combination of THC and CBD.

No signal could be detected for *Il17a* in male or female rats, either in their respective shams, in animals treated with VEH, or with any of the three cannabinoid oil regimens (Fig 4D).

### T cell-associated marker expression in male and female DRG

Of the markers that were initially examined in the thymus, only *Tnfa* and *Il17a* mRNA could be robustly and repeatably detected, and were therefore quantified relative to *βActin* in pooled L3-L6 DRG samples affected by the sciatic nerve cuff (Fig 4C and 4D, respectively). No signal was present for *Il2*, and while a signal for *Ifng* was present in the DRG of males and females, the very high Ct values precluded reliable quantitative analysis of this marker. No significant difference between male and female DRG shams was observed for either *Tnfa* (Fig 4C; P = 0.3166, t-test) or *Il17a* (Fi. 4D; P = 0.7065, t-test). While the male VEH-treated group trended to exhibit a significantly higher DRG expression levels of *Tnfa* relative to the male sham counterpart (Fig 4C), and this difference was significant for the male *Il17a* comparison (Fig 4D), no equivalent increase between sham and VEH was obtained in females. In addition, all three cannabinoid treatment regimens trended to increase the expression of endpoint levels of *Tnfa* (Fig 4C; male 1-way ANOVA: F = 10.44 and P = 0.0004) and *Il17a* (Fig 4D; male 1-way ANOVA: F = 11.53 and P = 0.0002) mRNAs in male DRG. In females, no significant differences occurred across any of the groups for either *Tnfa* (Fig 4C; female 1-way ANOVA: F = 1.436 and P = 0.2703) or *Il17a* (Fig 4D; female 1-way ANOVA: F = 0.9357 and P = 0.4683).

## Discussion

THC and CBD both play roles in modulating neurotransmission and immune function. When a nerve is injured or damaged, nociceptive pathways and the immune system are affected, and the dysregulation of these neuroimmune processes has been associated with pain chronification. By applying a preclinical model of peripheral NP based on long-term constriction of the sciatic nerve, the current investigation examined aspects of both processes in the context of whether treatment with naturally-derived cannabinoids could provide sustained analgesia several weeks after oral administration had ceased, and whether sex affected the outcome.

### Cannabinoid-mediated effects on mechanical nociceptive behaviour

The constriction model was behaviourally robust in males and females, as sham animals of each sex that underwent surgery without sciatic nerve cuff implantation did not exhibit, while those bearing a cuff and then orally treated with MCT oil alone, exhibited, mechanical hypersensitivity. Behavioural sex differences emerged upon oral gavage of natural cannabis extracts enriched for THC, CBD, or a 1:1 combination of THC and CBD. In males, all three 2-week cannabinoid treatment regimens protected the injured rats from developing significant mechanical hypersensitivity over time relative to vehicle-treated counterparts, as during this treatment window, none of these animals displayed paw withdrawal thresholds that differed significantly from pre-surgical behavioural profiles. In particular, the THC:CBD combination provided a beneficial effect in males one week after daily treatment ceased, which was

sustained until 7 weeks post-treatment. Of note, similar analgesic effects for THC and CBD were obtained by others using male Wistar rat chronic constriction injury models associated with significant mechanical hypersensitivity [55, 56]. Pure orally administered Δ9-THC (3 mg/kg in 10% cremophor; Sigma-Aldrich Chemie) attenuated nociceptive behaviour [56], and an evaluation of the therapeutic potential of purified CBD (2.5 to 20 mg/kg in methanol; GW Pharmaceuticals) administered via daily oral treatment from day 7 to 14 post-injury revealed a significant reduction in hypersensitivity to mechanical stimuli [55]. In females, none of the cannabis-derived oil formulations tested in the current investigation prevented the onset or further progression of mechanical hypersensitivity in response to sustained peripheral nerve constriction. Using the nerve cuff model, our group has observed a similar sexual dimorphism in behavioural treatment responsiveness to intraperitoneally injected cannabidiolic acid (CBDA)-methyl ester, a stabilized form of naturally derived CBDA that is a precursor of CBD [46].

## The effects of cannabinoid treatment on fiber-specific neurotransmission

With regard to neuronal activity, the threshold of unmyelinated small-diameter C HTM fibers was not affected by the natural cannabinoid extracts in either sex. C HTM and Aβ HTM neurons are classic nociceptors associated with pain, whereas Aβ LTM CUT and MS fibers become nociceptive in states of NP [40, 41], resulting in hypersensitivity. We have previously shown that the threshold of Aβ fibers is altered in the sciatic cuff model of NP, without any appreciable involvement of C fibers in either sex [57]. The DRG-recorded thresholds of male Aβ HTM, Aβ LTM CUT, and Aβ LTM MS neurons, which all represent large-diameter myelinated mechanoreceptive fibers, were significantly increased by early administration of THC as well as the 1:1 combination relative to vehicle, assessed seven weeks after discontinuing treatment. With the exception of THC significantly up-regulating the threshold of Aβ HTM neurons, none of the other myelinated mechanoreceptors were affected by any of the cannabinoid oil regimens in females. Others have shown that repeated early oral administration of rimonabant (SR141716), a selective CBR1 inverse agonist, in a male rat sciatic nerve constriction injury model substantially reduced the degeneration of myelinated fibers [58]. Costa et al. (2005) concluded that repair of the myelin sheath promoted sustained functional recovery assessed four weeks after treatment administration had ceased [58]. The sex differences in response to THC and the 1:1 combination of THC and CBD reported in the current investigation may be in line with peripheral myelin repair occurring only in males, warranting further investigation. Interestingly, oligodendrocytes and Schwann cells produce the myelin that enfolds neuronal axons (reviewed in [59]), with T lymphocytes context-dependently affecting this process [60–62]. In females, the immune system, which is thought to be basally "primed" for Th1 lymphocyte-mediated responses [18], may not support this putative type of repair, which would be in keeping with the recent suggestion that the higher prevalence of both autoimmune disorders and chronic neuropathic pain among women is more than coincidental (reviewed in [63]).

The ability to sense the position and movement of limb segments is required for maintaining balance, coordinated movement, and proper body orientation. MS fibers are considered to be the most important peripheral proprioceptor associated with these processes [64, 65]. There is also evidence supporting that cutaneous input [66] and joint receptors [67] contribute to proprioception, with an emerging improvement in the understanding of how central processing of relevant afferent signals changes during pain. In our model of NP, there was a clear sex difference in the thresholds of MS neurons, which have deep subcutaneous receptive fields and, unlike CUT neurons, are not responsive to cutaneous stimulation. Notably, while there were no statistically significant sex differences in the thresholds of CUT fibers assessed at the

DRG level, the responsiveness of MS fibers was distinct in males and females. The latter demonstrated a significantly sustained nerve cuff implantation-mediated drop in threshold relative to shams only in males, and MS neurons were also responsive to THC and the 1:1 combination treatment only in male rats. Interestingly, it has been proposed that sex differences related to neuromuscular control and proprioception [68–70] may contribute to differences in injury rates between men and women [71]. Given that THC affects aspects of proprioception in humans [72, 73], it will be of significant interest to further explore this finding in future experiments.

## Changes in immune marker profiles

Twice as many thymocytes are present in 3 month-old female rodents compared to age-matched males, and the thymic microenvironment is influenced by sex hormones [74]. The expression of markers associated with CD4+ and cytotoxic CD8+ T cells are basally higher in females (reviewed in [17]), while in males, testosterone serves as an immune modulator (reviewed in [75]) that suppresses CD4+ Th1 cell differentiation [76]. In the current investigation, we examined several markers associated with T lymphocytes at the mRNA levels, including *Il2*, *Ifng*, *Tnfa*, and *Il17a*. IL-2 is a tightly regulated cytokine produced by activated CD4 + and CD8+ T cells that context-dependently drives the thymic differentiation of T0 into Th1 or Th2 cells, or promotes the development of Treg cells. It has been linked to the chronication of pain [77, 78] and recently emerged as a potential pain biomarker in patients with sciatica [79]. IFN-γ is a well-known Th1 cell-produced cytokine, TNF-α is associated with a pro-inflammatory state and may also be secreted by Th1 cells [80], and IL-17A is a marker of CD4 + Th17 lymphocytes, which are associated with autoimmunity, inflammation, and chronic pain [81]. The levels of *Il2* and *Tnfa* mRNA were significantly higher in endpoint-collected thymi of female shams relative to male counterparts, supporting sex differences in the basal Th1 status. In response to peripheral nerve constriction, all of the selected markers underwent significant down-regulation in thymic mRNA levels in vehicle-treated males relative to shams, a profile that was restricted to *Il2* and *Tnfa* in the corresponding female comparison. This suggests that specific immunosuppression maybe occurring two months after inducing the peripheral nerve injury, at least at the level of T lymphocyte recruitment/ proliferation in the thymus.

We also examined whether there were any sustained cannabinoid-mediated changes in our immune marker repertoire relative to vehicle-treated males and females at experimental endpoint. Thymic *Il2* mRNA levels were significantly higher in treated males and females relative to vehicle, suggestive of sustained stimulation of T cell differentiation, irrespective of sex. Upon entering the splenic T-cell zone within the white pulp region, activated T cells have been shown to support increased expression of IL-2 [82], which has been linked to the development of autoimmunity [83, 84]. While the female splenic *Il2* mRNA profile reflected what was obtained in the thymus, the cannabinoid-mediated effects on the expression of this cytokine were sex-specific, suggesting that IL-2-mediated activity continued only in secondary female lymphoid tissues in response to each of the cannabinoid formulations. It is important note that, while numerous studies have shown cannabinoids to be largely immunosuppressive (reviewed in [26]), other research supports that CBD up-regulates the expression of IL-2 in female murine splenocytes [85–87]. Furthermore, it has been shown that CBD, by supporting IL-2 production under certain conditions, may contribute to a milieu that is required to drive the induction of Treg cells, demonstrating that CBD-mediated enhancement of a seemingly pro-inflammatory response may ultimately support immunosuppression [85]. Similarly, under some conditions, CBD either has no effect [88] or enhances its targeting [86] of IFN-γ.

In the latter scenario, as proposed by Kozela et al. (2016), CBD would increase the expression of IFN-γ-responsive genes, in turn attenuating T cell proliferation [88]. In this manner, although CBD appears to have "pro-inflammatory" effects on these particular cytokines, its consequence could overall result in an immunosuppressed state, which may be the case in females. Of interest, IL-2 may context-dependently block the differentiation of Th17 cells while also promoting their expansion [89]. This paradox is relevant, given that *Il17a* levels did not change in the female thymus in response to any of the cannabinoid treatments, although its expression was significantly up-regulated by all three oil formulations in males. In addition, in humans, TNF-α has been shown to assist in the efficiency of T0 cell differentiation into the Th17 lymphocyte subset [90], which is supported here by higher *Tnfa* levels in male thymi in response to early cannabinoid administration, an effect that did not occur in females. To our knowledge, there are no existing studies that have examined whether IL-17A expression is sex-dependently influenced by cannabinoid administration in models of NP, although there are reported sex differences in the murine Th17 axis in response to treatment with the PPAR-γ agonist pioglitazone [91].

Understanding the trafficking patterns of immune cells is important, as their function is influenced by tissue-specific microenvironments, with T cells in turn able to alter the cellular milieu of these niches [92, 93]. Changes in the periphery, including the presence of a nerve injury and the differentiation and recruitment of specific subsets of immune cells in lymphoid tissues, may influence the mobilization and trafficking of relevant cells not only to the site of damage, but also to affected DRG and into the CNS. A marked sexual dimorphism emerged upon examining the levels of *Tnfa* and *Il17a* in DRG associated with the injury-bearing limb. Compared to vehicle-treated animals, both of these cytokines were significantly up-regulated at the mRNA level in male DRG in response to each cannabinoid treatment regimen, an effect that again did not occur in females. These findings were unexpected, given that higher levels of TNF-α and IL-17A are generally associated with a pro-inflammatory state, and here, their increased expression was concomitant with sustained resolution of mechanical hypersensitivity and an increased activation threshold of myelinated Aβ mechanoreceptors. Interestingly, it has been reported that functional recovery after partial peripheral sciatic nerve ligation may require TNF-α, based on evidence that in knockout mice, nerve function remains impaired [94]. Although sex was not specified for the study conducted by Nadeau et al. (2011), it was concluded that neutralizing the actions of TNF-α most likely impairs peripheral nerve regeneration [94]. Furthermore, in a mouse chronic constriction injury model, IL-17A-positive Th17 cells were present in the endoneurium (the layer of connective tissue directly surrounding the myelin sheath of peripheral myelinated nerve fibers) associated with the damaged nerve [14], and cutaneous post-injury remodeling associated with a repair phenotype is mediated by IL17A-producing epidermal T cells [95].

## Summary

To summarize, it is possible that in male rats with a peripheral nerve constriction injury, cannabinoid treatment induces responses that promote repair by up-regulating specific "pro-inflammatory" immune cell populations. Through this process, THC and CBD may be promoting myelination, positively affecting the threshold of Aβ myelinated nociceptive fibers and returning associated nociceptive behaviours to baseline/sham levels. In contrast, in females, a pre-primed T lymphocyte program favouring increases in splenic IL-2 levels could supersede any beneficial effects induced by therapeutic cannabinoid administration, or excess splenic IL-2 may promote immunosuppression via Treg cells, thereby blocking nerve repair. Although not examined in the current investigation, the initial site of nerve injury may be important

when considering neuroimmune changes that occur in response to early treatment administration aimed at dampening central sensitization. In support of this notion, in a rat sciatic nerve injury model, it was shown that a large number of CD3+ cells (a marker of both CD4 + and CD8+ lymphocytes) were present around the injured section of the nerve three weeks post-surgery [96]. It will therefore be of considerable future interest to examine the area surrounding the injured site via a detailed time course analysis, facilitating an evaluation of whether sex-dependent nerve repair/regeneration or remyelination occur in cannabinoid-treated males and females relative to sham and vehicle-treated counterparts. In addition, it will be important to examine whether administering a higher oral dose of each cannabinoid, in oil form, to females could reverse mechanical hypersensitivity. However, we have shown that administering significantly higher concentrations of CBDA-methyl ester (CBDA-ME), a synthetic derivative of the naturally occurring precursor of CBD, to female rats implanted with a sciatic nerve cuff failed to reverse this response, with males undergoing complete recovery even at very low doses [46]. A plethora of research is now emerging in support of fundamental sex differences in nociceptive signaling, at least in response to a variety of treatment regimens including metformin [97] and, as recently reported by our group, pregabalin and progesterone [57]. The underlying basis of this dimorphic neuroimmune cross-talk may be rooted in hormonal and chromosomal differences in males and females.

## Conclusions

Oral administration of THC, CBD, and their 1:1 combination may be an effective treatment regimen to counter aberrant peripheral sensory activity and to modulate the immune response, especially when initiated at an early stage following peripheral nerve constriction. The application of these cannabinoids in males limits the development of persistent hypersensitivity. Additional research needs to be carried out to further assess whether cannabinoids elicit anti-nociceptive effects in females.

## Supporting information

**S1 Fig.** Thymic A) Il4 and B) Il10 mRNA levels were examined at experimental endpoint by RT-qPCR in males and females implanted with a sciatic nerve cuff that had undergone early cannabinoid treatment. Data are expressed as the mean ± SEM for each treatment group, and is presented relative to male sham-treated rats, with statistical significance analyzed by 1-way ANOVA with a Tukey's post-hoc comparison. Different letters (male comparison: a, b, etc.; female comparison: A, B, etc.) represent differences between groups (P<0.05). VEH: vehicle (MCT oil), Combo: 1:1 combination of THC and CBD.
(TIF)

## Author Contributions

**Conceptualization:** Katja Linher-Melville, Vikas Parihar, Ramesh Zacharias, Gurmit Singh.

**Formal analysis:** Katja Linher-Melville.

**Funding acquisition:** Katja Linher-Melville, Ramesh Zacharias, Gurmit Singh.

**Methodology:** Yong Fang Zhu, Jesse Sidhu, Natalka Parzei, Ayesha Shahid, Gireesh Seesankar, Danny Ma, Zhi Wang, Natalie Zacal, Manu Sharma.

**Project administration:** Katja Linher-Melville, Gurmit Singh.

**Resources:** Gurmit Singh.

**Supervision:** Katja Linher-Melville, Gurmit Singh.

**Validation:** Jesse Sidhu.

**Writing – original draft:** Katja Linher-Melville.

**Writing – review & editing:** Katja Linher-Melville, Yong Fang Zhu, Jesse Sidhu, Natalka Parzei, Gurmit Singh.

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
