## [Decision Letter · Decision Letter 0]

1 Apr 2020

PONE-D-20-02929

Evaluation of the preclinical analgesic efficacy of naturally derived, orally administered oil forms of Δ9-tetrahydrocannabinol (THC), cannabidiol (CBD), and their 1:1 combination

PLOS ONE

Dear Dr. Linher-Melville,

Thank you for submitting your manuscript to PLOS ONE. After careful consideration, we feel that it has merit but does not fully meet PLOS ONE’s publication criteria as it currently stands. Therefore, we invite you to submit a revised version of the manuscript that addresses the points raised during the review process.

Thank you for submitting this interesting study to evaluate sex differences in the effects of THC, CBD, and their combination in a rat model of peripheral neuropathic pain. Although no additional experiments will be required, it is imperative that you respond to each comment of both reviewers in your response letter, and refer each of these changes to the page numbers of the revised manuscript. Please note that all requested methodological details and statistical results are required, and the discussion section will require significant attention.

We would appreciate receiving your revised manuscript by May 16 2020 11:59PM. To enhance the reproducibility of your results, we recommend that if applicable you deposit your laboratory protocols in protocols.io, where a protocol can be assigned its own identifier (DOI) such that it can be cited independently in the future. For instructions see: http://journals.plos.org/plosone/s/submission-guidelines#loc-laboratory-protocols

We look forward to receiving your revised manuscript.

Kind regards,

Bradley Taylor

Academic Editor

PLOS ONE

Journal Requirements:

1. We note that you have included the phrase “data not shown” in your manuscript. Unfortunately, this does not meet our data sharing requirements. PLOS does not permit references to inaccessible data. We require that authors provide all relevant data within the paper, Supporting Information files, or in an acceptable, public repository. Please add a citation to support this phrase or upload the data that corresponds with these findings to a stable repository (such as Figshare or Dryad) and provide and URLs, DOIs, or accession numbers that may be used to access these data. Or, if the data are not a core part of the research being presented in your study, we ask that you remove the phrase that refers to these data.

2. Please amend your list of authors on the manuscript to ensure that each author is linked to an affiliation. Authors’ affiliations should reflect the institution where the work was done (if authors moved subsequently, you can also list the new affiliation stating “current affiliation:….” as necessary).

Reviewers' comments:

Reviewer's Responses to Questions

**Comments to the Author**

1. Is the manuscript technically sound, and do the data support the conclusions?

Reviewer #1: Yes

Reviewer #2: Partly

2. Has the statistical analysis been performed appropriately and rigorously? 

Reviewer #1: Yes

Reviewer #2: No

3. Have the authors made all data underlying the findings in their manuscript fully available?

Reviewer #1: Yes

Reviewer #2: No

4. Is the manuscript presented in an intelligible fashion and written in standard English?

Reviewer #1: Yes

Reviewer #2: No

5. Review Comments to the Author

Reviewer #1: The authors state that "In preclinical models representing surgically induced peripheral nerve damage, nociceptors undergo sensitization, with afferent fibres developing patterns of ectopic neural discharge." This is not always true see ectopic discharges from injured nerve fibers are highly correlated with tactile allodynia only in early, but not late, stage in rats with spinal nerve ligation. Sun et al 2005.

Authors state that "Initiating treatment at an early stage post-nerve injury (for example, PRIOR TO or directly after a

surgery" but they provide no evidence for treatment helping before. The citation only includes early intervention. If NP is caused by by aberrant neural discharge what can be done to prevent it from developing?

Authors state "The predominant components of phytocannabinoid extracts, Δ9-tetrahydrocannabinol (THC) and cannabidiol (CBD), are able to modulate nociceptive thresholds and influence the release of pro-inflammatory molecules and chemokines by relevant non-neuronal cell types [reviewed in (18)]." The citation refers to an entire edition of a journal a more specific citation is needed for a direct claim.

Authors do not mention CBDs direct agonism of 5HT1A which seems like a significant oversight

Authors do not justify the use of natural cannabis abstracts, which contain impurities. Why not use controlled synthetically produced compounds?

Authors have not provided any evidence or experiments to determine what THC and CBD are doing to produce the observed effects. Antagonist pharmacology to determine the exact nature of the effects of THC and CBD would significantly help the central argument of the paper.

Reviewer #2: Overall Evaluation

This is an interesting study to evaluate the effects of THC, CBD, and their combination in a rat model of peripheral neuropathic pain. The large sex differences are particularly interesting, but shortcomings in description of the Methods and Results, together with an unfocused discussion that does not adequately reference the existing literature on sex differences in the effects of cannabinoids in rodent models of neuropathic pain, must be addressed before this can be further considered for publication. Numerous additional minor comments are also interwoven with major comments throughout the following critique.

Critique

1. The Methods are missing numerous key details.

a. The cuff model is generally referred to as a preclinical model of peripheral neuropathic pain and not a preclinical model of “post-surgical neuropathic pain.” Otherwise, all of the preclinical models of PNP would have been referred to as models of “post-surgical neuropathic pain.”

b. Detailed surgical methods are required. Was anesthesia used? Post-operative analgesics? Details of the cuff including material and size are needed. More fully describe the sham surgery, did it include isolation of the nerve?.

c. Literature reference to the cuff model is needed in Methods

d. Details of the feeding needles is needed in Methods. What was volume range of oral administration?

e. Please explain why only vF>9 was chosen as characteristic of baseline, (why not vF>8 for example?), and list numbers of animals removed from the study based on this criteria.

f. What is “responsive males” on page 8? Please indicate whether any animals were excluded for any reason after surgery

g. Please indicate source vF filaments, indicate which ones were used, and briefly describe the up-down method used. Clarify the statement “Data reflecting the paw withdrawal force normalized ot baseline”. Data is and should be plotted in gram force. “paw withdrawal force” is cumbersome.

h. At what time of day were behavioral experiments performed? By which experimenter? Ddi the “dedicated research staff” who handled the animals vary from day to day? Was the experimenter blind to treatment and if so what was the blinding procedure.

i. Section on in vivo intracellular DRG recordings requires description of animal preparation including post-surgery date, anesthesia, surgery, and monitoring. Description of electrical stimulation are required throughout this section.

j. Criteria for fiber classification are required.

k. Description of stimuli including ‘muscle belly’ and ‘cutaneous’ is required

2. The Results section

a. It is controversial to use the term “chronic” for a behavior lasting a small number of weeks. Same for the term “allodynia” which may or may not be reflected by hypersensitivity to vF filaments. Use correct terminology, for example change “chronic mechanical allodynia” to “persistent mechanical hypersensitivity”

b. An appropriate rationale for choosing to focus on Week 3 and Week 6 would be helpful. Should definitely NOT have been chosen to subjectively determine any greater effect of one drug / drug combination over another. Alternative is to display area under the curve across weeks, or averages across weeks.

c. When statistically significant, F and P values of the 2-way ANOVA’s are required.

d. The manuscript Results must be very clear to state that the data of Fig 1 and Fig 2 are the same data, just plotted differently. This should also be stated in the figure caption to Fig 2.

e. Much of the information in the Figure Captions belongs in the Methods.

f. For Fig 3, “CUT” vs not cut neurons are not defined. Explain why only delineated for A-beta? Were CUT neurons sensitized as compare to “MS” neurons – statistical comparison may be useful here. Discussion should compare this difference to the literature.

g. Fig 3A does not make sense. For example, 4000 pA amplitude is not shown. The authors must explicitly state which groups are different, as designated by upper or lower case letters. Difference from vehicle? Again, statistics results of 2-way ANOVA are required.

h. The data of Figure would be much more impactful if data SHAM vs INJURED are shown. Why not? If undetectable levels in SHAM then this should be reported. It is very important to show whether injury changes any of these T cell markers, and to discuss whether any such change is reversed or enhanced by the drugs. Again, F and p value results of ANOVA are required in the Results section.

3. The Discussion requires organization. Use informative subheadings. One key conclusion per paragraph, please. The discussion of T-cells is far too long and speculative.

a. Line 1 of the Discussion seems to state that THC and CBD are associated with pain chronification. Isn’t it the opposite?

b. p.23 discussion of neural repair is much too lengthy in the absence of any data on neural repair.

c. P.24 indicates that the literature shows higher levels of T cells in females. This contradicts most of the data shown in Figure 4. The authors speculate this is due to the nerve injury – the authors ran Sham injured rats and should compare SHAM vs CUT data to address their speculation.

4. The Discussion must relate the findings of Figs 1-3 to the existing literature. This must be done thoroughly, with greatest attention to the strongest findings.

6. PLOS authors have the option to publish the peer review history of their article (what does this mean?). If published, this will include your full peer review and any attached files.

Reviewer #1: No

Reviewer #2: No

---

## [Author Response · Author response to Decision Letter 0]

15 Apr 2020

Please refer to the attached file labeled as "Response to Reviewers", which includes a detailed point-by-point of each of the editorial changes required, as well as a redress of each of the reviewers' comments. Thank you.

---

## [Decision Letter · Decision Letter 1]

21 May 2020

Evaluation of the preclinical analgesic efficacy of naturally derived, orally administered oil forms of Δ9-tetrahydrocannabinol (THC), cannabidiol (CBD), and their 1:1 combination

PONE-D-20-02929R1

Dear Dr. Linher-Melville,

We are pleased to inform you that your manuscript has been judged scientifically suitable for publication and will be formally accepted for publication once it complies with all outstanding technical requirements.

With kind regards,

Bradley Taylor

Academic Editor

PLOS ONE

Additional Editor Comments (optional):

A much improved revision that rigorously addresses all comments of both reviewers.

Reviewers' comments:

Reviewer's Responses to Questions

**Comments to the Author**

1. If the authors have adequately addressed your comments raised in a previous round of review and you feel that this manuscript is now acceptable for publication, you may indicate that here to bypass the “Comments to the Author” section, enter your conflict of interest statement in the “Confidential to Editor” section, and submit your "Accept" recommendation.

Reviewer #1: (No Response)

2. Is the manuscript technically sound, and do the data support the conclusions?

Reviewer #1: Yes

3. Has the statistical analysis been performed appropriately and rigorously? 

Reviewer #1: Yes

4. Have the authors made all data underlying the findings in their manuscript fully available?

Reviewer #1: Yes

5. Is the manuscript presented in an intelligible fashion and written in standard English?

Reviewer #1: Yes

6. Review Comments to the Author

Reviewer #1: My comments were partially addressed. The minor changes were made although there was no addition of more conclusive pharmacology to determine the how THC and CBD are causing these effects. I think this would greatly increase the impact of the data but the current data presented is technically sound.

7. PLOS authors have the option to publish the peer review history of their article (what does this mean?). If published, this will include your full peer review and any attached files.

Reviewer #1: No

---

## [Editor Report · Acceptance letter]

26 May 2020

PONE-D-20-02929R1 

Evaluation of the preclinical analgesic efficacy of naturally derived, orally administered oil forms of Δ9-tetrahydrocannabinol (THC), cannabidiol (CBD), and their 1:1 combination 

Dear Dr. Linher-Melville:

I am pleased to inform you that your manuscript has been deemed suitable for publication in PLOS ONE. Congratulations! Your manuscript is now with our production department. 

With kind regards,

on behalf of

Dr. Bradley Taylor 

Academic Editor

PLOS ONE